



**Pacific Decadal Oscillation and recent oxygen decline in the eastern tropical Pacific Ocean**

Olaf Duteil[1], Andreas Oschlies[1], Claus W. Böning[1]

[1] GEOMAR – Helmholtz Centre for Ocean Research Kiel, Düsternbrooker Weg. 20, 24103 Kiel, Germany

*Correspondence to* **:** Olaf Duteil (oduteil@geomar.de)

**Abstract**

The impact of the positive and negative phases of the Pacific Decadal Oscillation (PDO) on the extension of the poorly oxygenated regions of the eastern Pacific ocean has been assessed using a coupled ocean circulation-biogeochemical model. We show that during a "typical" PDO positive phase the volume of the suboxic regions expends by 7 % in 50 years due to a slow-down of the large scale circulation related with the decrease of the intensity of the trade winds. The oxygen

levels are mostly constrained by advective processes between 10°N and 10°S while the diffusive processes are dominant poleward of 10°: in a "typical" PDO positive phase the sluggish equatorial current system provides less oxygen into the eastern equatorial part of the basin while the oxygen transport by diffusive processes significantly decreases south of 10°S. The region located north of 10°N displays less sensitivity to the phase of the PDO as the local upwelling-related processes play

a dominant role compared to the large scale circulation in setting the oxygen concentration. Our study suggests that the prevailing PDO positive conditions since 1975 may explain a significant part of the current deoxygenation occurring in the eastern Pacific Ocean.

**1. Introduction**

Oxygen is one of the most important chemical elements in the ocean, as marine organisms ranging from microorganisms to vertebrates use it for respiration. Its concentration is regulated both by circulation and by biogeochemical processes. Observations provide a picture of the mean oxygen distribution: high latitude regions are characterized by high oxygen concentrations, while

subsurface tropical regions are poorly oxygenated. In tropical regions, the large export of organic material combined with the sluggish circulation depletes oxygen levels at depth, resulting in the formation of large suboxic regions where the oxygen concentration falls below 20 mmol.m$^{-3}$ (Karstensen et al., 2008). Ocean models capture this mean picture reasonably well (Bopp et al., 2013; Cocco et al., 2013, Cabré et al., 2015, Shigemitsu et al., 2017).





The temporal variability of oxygen concentration in the interior ocean is nevertheless still poorly understood, in particular in the suboxic regions of the eastern tropical Pacific Ocean, where oxygen concentrations are among the lowest worldwide (Karstensen et al., 2008). A strong decrease of oxygen concentrations in these regions has been inferred from observations from the 1960s to the

2000s (Stramma et al., 2008, 2012; Schmidtko et al., 2017). This deoxygenation is not correctly reproduced by ocean models (Oschlies et al., 2017). It is not clear whether the changes in oxygen concentration are due to anthropogenic changes (Matear et al., 2003; Long et al., 2016, Ito et al., 2017), or are related to natural low-frequency climate oscillations such as the Pacific Decadal Oscillation (PDO) (Deutsch et al., 2014), a robust and recurring pattern of ocean atmosphere

climate variability centered over the mid-latitude north Pacific basin (Mantua et al., 1997). Could the shift from a negative phase of the PDO (prevailing conditions before 1975) to a positive phase (prevailing conditions since 1975) partly explain the current deoxygenation occurring in the eastern tropical Pacific Ocean ? Which mechanisms are at play and in which regions ?

The stronger trade winds occurring during a negative phase of the PDO causes a shoaling of the eastern thermocline of the tropical and subtropical Pacific Ocean (Miller et al., 1994). Deutsch et al. (2011, 2014) showed that the depth of the thermocline regulates the oxygen levels in the coastal regions of the north eastern subtropical Pacific Ocean. In these regions, a shallower thermocline fosters low oxygen concentration in the intermediate ocean as a larger amount of organic material is

respired below the mixed layer (Deutsch et al., 2011). Simultaneously, in the tropical regions the zonal volume transport by the equatorial current system increases during a negative PDO event as well as the meridional transport by the sub-tropical cells (STC) (Hong et al., 2014) which connect the subtropics to the tropics (McCreary and Lu, 1994). The variability of the strength of the STC forces the variability of the oxygen transport in the upper thermocline of the equatorial Pacific

Ocean (Duteil et al., 2014a). A competition takes place between the increased oxygen transport and the increased respiration as primary production is fueled by the increased nutrient supply. Finally, stronger trade winds also increase the subduction volume of the North (Qu et al., 2009) and South Pacific Eastern Subtropical Mode Water (Luo et al., 2011). A subtropical increase of productivity related with an increase of the trade winds causes a negative oxygen anomaly in these mode water,

which is transported equatorward and leads to a delayed oxygen decrease in tropical regions as shown by Ridder and England (2014) in an Earth System Model.



All these studies highlight the potential effect of the PDO on oxygen concentration and show that the processes at play are diverse and strongly region dependent. However, we still do not have a clear picture of the impact of the PDO on the suboxic regions. Indeed the studies cited above focus either on the north east Pacific coastal regions (Deutsch et al., 2011, 2014) or on the upper thermocline of the tropical Pacific Ocean (Duteil et al., 2014a). A caveat of the model used by Ridder and England (2014) is that it represents poorly the equatorial undercurrent (EUC), as most of the coarse (resolution lower than 0.5° at the equator) models (Karnauskas et al., 2012). The EUC indeed ventilates the suboxic regions, as shown by Cabre et al. (2015), Shigemitsu et al. (2017) in a range of models part of the Coupled Model Intercomparison Project 5 (CMIP5).

Understanding precisely the role of the PDO in setting oxygen levels in suboxic regions is difficult in "traditional" 50-years hindcasts experiments (Deutsch et al.,2011; Ito et al., 2013; Duteil et al., 2014) due to the superimposition of a long term climate trend and higher frequency climate oscillations such as the El Nino Southern Oscillation (ENSO) (Ito et al., 2013, 2016; Eddebbar et al., 2017). These time scales of variability may interact. Ito et al. (2013) showed that the temporal spectrum of oxygen concentration of the north east tropical Pacific Ocean is characterized by a strong decadal variance which may partly arise from the 'reddening' of the variability spectrum of the physical and biological drivers (Ito and Deutsch, 2010), suggesting a contribution of ENSO to the oxygen decadal variability, in complement to the PDO.

Here rather than performing an "hindcast" experiment, we assess specifically the role of the phase of the PDO on the suboxic regions of the tropical eastern Pacific Ocean by forcing a coupled circulation – biogeochemical model using "typical" conditions characteristic of the negative and positive PDO phase. These reconstructed atmospheric forcings are derived from monthly averages of realistic winds and heat fluxes of the 1948 – 2007 COREv2 dataset (Large and Yeager, 2009). Our aim is to understand whether and by which processes (oxygen advection, diffusion, respiration) the PDO may be responsible of the observed oxygen decline in the suboxic regions of the tropical eastern Pacific Ocean.

This paper is organized as follows. The section 2 details how the "typical" PDO forcings have been constructed and the experiments that we perform. In section 3, we assess the basin scale circulation of our experiments. In section 4, we present the difference in oxygen levels between a "typical" positive and negative PDO phase. The mechanisms regulating the oxygen levels are described in



section 5. Temporal aspects are discussed in Section 6. In section 7 we discuss the changes in the upwelling systems and the impact on suboxia. We summarize our results in section 8.

**2. Forcings and experiments**

The NEMO ocean model version v3.6 has been used (Madec et al., 2008) in the standard configuration ORCA2. This configuration has been widely used in previous studies and constitutes the ocean component of the ISPL-CM5A model, part of the Coupled Model Intercomparison Project (CMIP5) effort (Dufresnes et al., 2013). Its zonal resolution is 2 degrees. The mean meridional resolution is 2 degrees outside the tropics and increases to 0.5 degrees close to the equator. The

resolution of ORCA2 is sufficient to reproduce realistically the EUC (Cravatte et al., 2007) and the subtropics - tropics connectivity (Luebbecke et al., 2008). The circulation model has been coupled to a 6-compartments (Nutrient, Phytoplankton, Zooplankton, Particulates and dissolved detritus, oxygen) biogeochemical model . This model is described in detail in Kriest et al. (2010). It has been adapted by Duteil et al. (2014a,b) to the NEMO framework.


We constructed 3 atmospheric forcings datasets derived from the interannual 1948 – 2007 COREv2, 6h temporal resolution, forcing dataset (Large and Yeager, 2009):

- MEAN : 1- A low-pass filter has been applied to remove the frequencies with a period shorter than 1 month. 2- The long term trend 1948 – 2007 (Yang et al., 2016) has been removed. 3- The

corresponding time steps of the individual annual forcings of the year 1948 – 2007 have been averaged, leading to the reconstruction of a 1 year, 6h temporal resolution, climatological forcing. The difference between MEAN and the COREv2 "normal year" is the absence of high frequency variability ( < 1 month) and the removal of the long term trend implicitly contained in the "normal year"

- WARM (COLD) : the steps 1- and 2- are similar as above. In WARM (COLD), the corresponding time step of the forcings characterized by a positive (negative) PDO phase (Fig. 1a) have been averaged, leading to the reconstruction of a 1 year, 6h temporal resolution, climatological forcing "typical" of a PDO positive (negative) phase.

In WARM the zonal wind speed decreases by about 0.2 to 0.5 ms-1 compared to MEAN in the mid-equatorial Pacific Ocean, where the winds are strongest (at least 8 ms-1) (Fig. 1b). It increases close to the eastern coast by up to 0.3 ms-1, where the winds are weaker (2 to 8 ms-1). A similar pattern has been described by Merrifield et al. (2012) and Zhou et al. (2017). The meridional wind speed




decreases by about 0.2 ms-1 (Fig. 1c). The 10m air temperature increases by 0.1 to 0.3 °C in the

eastern Pacific Ocean and decreases in the gyres (Fig. 1d). COLD presents the opposite pattern.

We spin up the model during 1000 years using the forcing dataset MEAN. We subsequently performed 2 experiments that we integrate for a period of 50 years, which corresponds to the typical oscillation period of the PDO during the past 200 years (Mc Donald and Case, 2005).

-WARM: WARM forcing set (zonal and meridional wind speed, 10m air temperature, 10m humidity )

-COLD : COLD forcing set (zonal and meridional wind speed, 10m air temperature, 10m humidity)

## 3. Basin scale circulation

### 3.1. Gyres

The subtropical gyre (STG) slows down and extends equatorward in WARM, constraining the tropical gyres (TG) (Fig 1e). The slow-down reaches up to 5 Sv (or 5-10 %). More particularly the almost zonal boundary between the northern STG and the TG is shifted at 10°N by 1-2° of latitude. Incursions of the southern STG into the TG are identified at 100°W/10°S. This picture is coherent

with an analysis of the Simple Ocean Data Assimilation experiment which shows that the northern subtropical gyre was weaker and to the north before 1976–1977 (negative PDO phase), and stronger and to the south after 1976–1977 (positive PDO phase) (Jiang et al., 2013) . The thermocline depth shallows in WARM in the STG and deepens in the eastern tropical part of the basin (Fig 1f). The large signal observed at 10°N is likely related with the extension of the STG. The passive

adjustment time of the ocean (without considering ocean atmosphere feedbacks) is quick (a few years), which is coherent with previous studies (Zhang and Delworth, 2015; Deser et al., 1999; Hong et al., 2014).

### 3.2. Meridional overturning

The upper meridional overturning is characterized by the presence of the Subtropical-Tropical Cells (STC) (Fig 1g). These cells are shallow (upper 500 m meters) structures connecting the subtropics and tropical regions (McCreary and Lu, 1994) and respond to a change in wind stress by baroclinic adjustment (Hong et al., 2014). The strength of the STCs (and therefore of the whole tropical current system, including the equatorial upwelling and the equatorial undercurrent) decreases by up

to 5 Sv (10%) in WARM compared to COLD. The order of magnitude of the strength of the STCs in the WARM and COLD experiments are in line with other modeling studies (Lohman and Latif, 2005; Luebbecke et al., 2008; Hong et al., 2014) and with the observational study of McPhaden and





Zhang (2002), who showed that the equatorial upwelling decreased by 10 % from 1970-77 (47 Sv) to 1980-89 (42 Sv) related with a shift of the phase of the PDO.


## 4. Oxygen concentration

### 4.1. Comparison with the World Ocean Atlas

At the end of the spinup, the model reproduces the large-scale features of the observed World Ocean Atlas (WOA) (Garcia et al., 2013) oxygen concentration field (Fig. 2a). The thickness of the SUB20

regions, defined as the regions where oxygen concentrations are lower than 20 mmol.m$^{-3}$ at the end of the spinup, reaches more than 700 m north of the equator both in the WOA (Fig. 2b) and in the model (Fig. 2c). 'Typical' biases (Bopp et al., 2013; Cabre et al., 2015) are present in our model. In particular: 1- the OMZ region does not extend far enough westward, in particular north of the equator, 2- the concentration at the equator is too low, maybe due to a poor representation of the

intermediate current system, located below the EUC, in relatively coarse resolution models (Marin et al., 2010; Getzlaff and Dietze, 2013). The thickness of the suboxic regions is nevertheless lower in the equatorial region compared to the tropics, as shown in Fig. 2c

### 4.2. Perturbation by the PDO

After 50 years of integration, the oxygen (average 100 – 700 m) concentration is lower in the eastern part of the basin in WARM compared to COLD (Fig. 2d). This decreases reaches up to 100 % in regions where the oxygen is very low (below 5 mmol.m$^{-3}$) and about 5-10 % in regions where the oxygen concentration is lower than about 20 mmol.m$^{-3}$ (Fig. 2e). The  volume of the SUB20 regions is larger by 7 % in WARM and the thickness of the suboxic layer increases by up to 100 m

close to the coast between 10°N and 10°S and at the outer boundary of the SUB20 regions (Fig. 2f) (note that the "stepwise shape" of the anomaly is due to the discretization of the vertical grid of the ocean model). The oxygen concentration in the SUB20 regions decreases by 2-10 mmol.m$^{-3}$ in WARM compared to COLD (0.04 to 0.2 mmol.m$^{-3}$.yr$^{-1}$). Conversely, in the mid-Pacific Ocean oxygen concentrations are larger in COLD by 2 to 20 %. This increase is localized (5-10°N and 5-

10°S and eastward of 160°W).

Our results can be put in perspective with observations. An oxygen decrease by 1 mmol.m$^{-3}$yr$^{-1}$ has been monitored in the eastern equatorial region (85°W, 2°S to 8°S) since 1976 (Czeschel et al., 2012). Schmidtko et al. (2017) found a global decrease of the integrated oxygen concentration in

the water column since 1960. This decrease is of the order of 0.2 mmol.m$^{-3}$yr$^{-1}$ in the equatorial Pacific ocean at 300 m depth. Similarly, Ito et al. (2017) shows that oxygen declines at 400 m depth



by 0.2 mmol.m$^{-3}$ .yr$^{-1}$ since 1958 in the eastern Pacific Ocean; at 100 m depth, oxygen decreases by up to 0.4 mmol.m$^{-3}$yr$^{-1}$ . They however observed a localized oxygen increase at 10°N, similar to the one that we described above. Our simulations suggest that a shift from a negative to a positive phase

of the PDO may be responsible of a large fraction of the observed oxygen decrease.

### 5. Regulation of the oxygen levels

The oxygen level below the euphotic zone is determined by the balance between consumption (respiration) and supply (transport). The supply is decomposed into advective, diapycnal and

isopycnal diffusion terms. The analysis is based on the average of the 50 years of integration.

### 5.1. Intermediate (100 – 700m) tropical Pacific Ocean

The respiration processes remove oxygen (Fig. 3a), especially in the tropical regions (up to 10mmol.m$^{-3}$yr$^{-1}$ in the layer 100-700 m ), as the biological production is high along the equator and

close to the coast. In the experiment WARM the basin-scale circulation is more sluggish than in COLD (See 3. 'Basin Scale Circulation'), leading to a decrease of the concentration of nutrients in the euphotic zone (see '7. Productivity and Upwellings'). The respiration term becomes "less negative" in WARM (positive anomaly of 1-2 mmol.m$^{-3}$ yr$^{-1}$ in the equatorial region - Fig. 2d).

The removal of oxygen is compensated by supply processes (Fig. 3b), partly performed by advective processes (Fig. 3c) which dominate the supply budget between 5°N and 5°S and in the eastern part of the basin, showing the preponderant role of the equatorial current system to supply oxygen to the oxygen-depleted regions (Cabre et al., 2015; Shigemitsu et al., 2017). Conversely the diffusive processes (isopycnal and diapycnal mixing) dominate the supply budget outside of the

equatorial region (poleward of 10°). (Fig. 3d). The ocean currents shape the thermocline and high oxygen concentrations are transferred by isopycnal diffusion from the core of the EUC to the poleward intermediate ocean.

The WARM – COLD oxygen anomaly mirrors the changes in respiration in the western part of the

basin. The decrease of the supply term (particularly the advective terms) is stronger than in the eastern part (east of 100°W), leading to the decrease of the oxygen levels.

### 5.2. SUB20 regions

The contribution of each process has been vertically and zonally averaged over the SUB20 region

of the Pacific Ocean (Fig. 4a) and multiplied by the longitudinal extension of the SUB20 of the

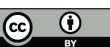



experiment COLD. The total supply term (Fig. 4a - black) is characterized by a large supply in the equatorial region, between 10°N and 10°S, due to advective processes (Fig. 4a - red). The role of the westward South and North equatorial currents are clearly apparent at 5°N and 5°S. At this location the thermocline exhibits a strong slope fostering isopycnal diffusion (Fig. 4a - blue),

removing oxygen from the equator (and the EUC) and transferring it to the deeper, adjacent regions. The effect of the jets and the isopycnal diffusion are adding up and cause the strong peak in oxygen supply, located between 5°S and 10°S (and to a lesser extent between 5°N and 10°N). The role of the oxygen supply by diapycnal diffusion (Fig 4a. – green) is relatively small in the equatorial region (about 20% of the total supply), but more significant between 30°S and 10°S (about one third

of the total supply) and dominant north of 10°N (Fig 4b). Between 10°S and 30°S, the isopycnal diffusion term (Fig. 4a – blue) plays a dominant role, possibly due the outcrop of isopycnals and the formation of the mode water close to the southern part of SUB20. The importance of the isopycnal diffusion in setting the oxygen levels in the region off Chile (around 30°S) has been previously highlighted by sensitivity tests to the Redi mixing coefficient (Gnanadesikan et al. 2012). Stramma

et al. (2010) roughly estimated the oxygen budget (30°N – 30°S) in the suboxic regions based on observational data. Despite large uncertainties, it points to an allocation of about 33 % by advection, 22 % by vertical mixing and 45 % by eddy mixing (Brandt et al., 2015). In our model, averaging the SUB20 budget between 30°N and 30°S gives comparable size order (Fig. 4b) (in COLD: 21 % of the supply occurs by advection, 29 % by diapycnal mixing and 50 % by isopycnal mixing, strongly

related with the mesosocale activity).

In the experiment WARM, the supply of $O_2$ by circulation processes decreases (Fig. 4c – black) due to a reduction of the advective supply in the equatorial region (Fig. 4d – red) and of the diapycnal and isopycnal diffusion poleward of 10°N and 10°S (Fig. 4d – blue and green). The primary

production decreases as well, resulting in a positive anomaly of the respiration term (which is 'less negative') (Fig. 4c – green). The decrease of respiration nearly compensate the reduced supply of oxygen especially in the equatorial region: the decrease of the circulation supply terms are however larger, leading to a net decrease of oxygen levels in SUB20 (Fig. 4c – red). In the region 10°N-30°N, about half of the oxygen decrease is caused by changes of the supply, while the other half is

due to changes in respiration, triggered by changes in the advective processes (see 7 'Productivity and upwellings'). In the region 10°S-30°S, the decrease of the oxygen levels is principally due to a decrease of the isopycnal and mixing processes (Fig. 4d). More generally, the relative importance of diapycnal diffusion increases in the WARM experiment compared to COLD (20 % of the supply occurs by advection,  27 % by diapycnal mixing and 53 % by isopycnal mixing) (Fig. 4b)





## 6. Temporal aspects

### 6.1. Intermediate (100 – 700m) tropical Pacific Ocean

In the western part of the basin (160°E-140°W, 10°N-10°S), oxygen concentration decreases in WARM compared to COLD in the first years of the experiment (initial shock, highlighting the time

scale of response of productivity to a change in circulation) and increases afterwards (Fig 5.a-d). The changes in the advective processes (Fig 5e. - blue) are largely responsible of the total changes in the supply (Fig. 5e - bold black). The decrease in respiration (Fig 6e - green) offsets these changes, leading to net a positive anomaly (Fig 5e - bold red). After 50 years the positive anomaly is still growing but at a slower pace (Fig. 5a): diffusive processes (cyan) (and more specifically

diapycnal diffusion (pink) act in the same direction as the currents-driven changes as a result of a less stratified upper ocean under WARM conditions.

### 6.2 SUB20 regions

The picture described above shows similarities with the one of the equatorial region (10°N-10°S) of

SUB20, where a strong decrease of the oxygen supply (Fig. 5f – bold black) due to advective processes (Fig. 5f – blue) occurs. The role of diffusion (Fig. 5f – pink) is however larger in SUB20 compared to the mid Pacific equatorial region. Respiration changes (Fig. 5f – green) do not offset the weaker supply, as in the western tropical Pacific Ocean, leading to a net decrease of oxygen concentration (Fig 5a-d). In the northern part (10°N-30°N) of SUB20 an increase of oxygen

transport by advective processes (Fig. 5g - blue) is compensated by a strong decrease by diffusion processes (less mixing), leading to a net decrease of oxygen supply. Primary production and respiration increases (see 7 'Productivity and upwellings'), reinforcing the decrease of oxygen levels. In the southern part (10°S-30°S) of SUB20 the decrease of the oxygen supply by advective transport (Fig 5h – blue) is accompanied by a strong decrease of supply by diapycnal (Fig. 6h –

pink) and isopycnal mixing (Fig. 5h - cyan). The adjustment occurs at (multi) decadal time scale (Fig. 5a-d). The strong changes in mixing in the north / south part of the SUB20 regions suggest a stronger influence of the subtropical regime in WARM than in COLD in SUB20 (see 3. 'Basin Scale Circulation'), which explains the 'initial shock' related with a change of regime.

The simulated PDO-induced changes are significant after at least 5 years (in the equatorial Pacific ocean) to a few decades (in the Southern part of the suboxic regions). Significant changes occurred after a similar time scale in the study of Ridder and England, 2014. It suggests that the role of higher frequency climate oscillations such as the El Nino Southern Oscillation (ENSO) – or "short-





lasting" PDO events - have a very limited impact on oxygen concentrations of the suboxic ocean, in

agreement with Deutsch et al. (2011) and Ito et al. (2013). ENSO may however have an impact on
the surface air/sea oxygen exchanges (Eddebar et al., 2017) and "short-lasting" PDO event (less
than 10 years) on the oxygen concentration of the upper thermocline of the mid-Pacific Ocean
(Duteil et al., 2014b). In the southern region (10°S-30°S), the system is still losing oxygen after 50
years of integration suggesting that extra-tropical processes are involved. Getzlaff et al.. (2016)

showed that a vigorous subtropical gyre, driven by an increases of the westerlies, supplies oxygen
to the tropics. Yamamoto et al. (2015), Keller et al. (2016) showed that high latitudes may constrain
tropical suboxic regions at multi-decadal and centennial time scale.

## 7. Productivity and upwellings

As we have seen before the changes in respiration play a significant role in setting the oxygen
levels. Respiration either compensates the changes in supply in the equatorial region or acts in
synergy with the decrease in supply to deplete oxygen in the northern part of SUB20 in the WARM
experiment. While the changes in oxygen supply and transport are primarily linked with changes of
the structure of the interior ocean, the change in respiration is primarily linked with the surface and

upper thermocline productivity, which ultimately depends of the supply of nutrients to the mixed
layer.

### 7.1. Upwellings strength and seasonality

In the WARM experiment, the nutrient supply decreases over most of the basin, leading to a

decrease of the nutrient uptake and the productivity. The decrease of the supply is caused both by a
slowing-down of the circulation and by a thermocline deepening (as seen in 3. "Basin Scale
Circulation"). However, counter intuitively, the nutrients levels increase and the production is
slightly (2-5%) stimulated in WARM in the eastern upwelling systems. The Fig. 6a present
similarities with the imprint of the PDO on deseasonalized chlorophyll concentration inferred from

satellite data (Thomas et al., 2012). They found that a positive phase of the PDO is associated with
a general decrease of chlorophyll concentration in the tropical Pacific Ocean. In the region located
between 15°N-30°N and east of 140°W, and in the region south of 10°S and east of 120°W the
correlation between PDO and deseasonalized chlorophyll is however positive (their figure 7).
Furthermore other climate oscillations, such as the North Pacific Gyre Oscillation (NPGO)

constrain the strength of the upwelling cells in the ETNP in complement to the PDO (DiLorenzo et
al., 2008; Macias et al., 2012).





This stimulation is due to change of the seasonality of the upwelling system in our experiments. The Eastern Tropical South Pacific (ETSP) region (average 10°S-30°S, 90°W-60°W) is characterized a

strong downwelling in May / April and a strong upwelling in August / September in COLD (Fig. 6c). Conversely, in WARM, the ETSP is characterized by a weaker upwelling which albeit lasts during all the year, continuously supplying the ML and the upper ocean with nutrients and creating a positive anomaly in nutrients and productivity (Fig 6e and Fig 6g). In the Eastern Tropical North Pacific (ETNP) (10°N-30°N, 120°W-90°W), a strong upwelling occurs one month early in WARM

than in in COLD (Fig. 6d), leading to an increase in nutrient concentrations in the upper ocean (Fig. 6f and Fig. 6h) in a season where irradiance is high, fostering primary production. The positive PDO years used to construct the WARM forcing are constituted by 30 % of positive NPGO years, which may explain the shift in the upwelling seasonality and the stimulation of productivity in the ETNP (Chenillat et al., 2012)


**7.2. Role of local vs large scale circulation**

In order to disentangle the changes of the local forcings, related with the upwelling systems, and the remote forcings (trade winds) on the eastern oxygen levels and productivity, we perform 2 supplementary experiments, COLD50 and WARM50. These experiments are similar to COLD and

WARM. However, where oxygen is, anywhere in the water column and for the respective month, lower than 50 mmol.m$^{-3}$ the MEAN surface is employed (see 2 – "forcing and experiments").

The Sea Surface Height difference and the changes in circulation between WARM50 and COLD50 display strong similarities than between WARM and COLD. The amplitude is however weaker in

the eastern part of the Pacific basin; the equatorward shift of the subtropical gyres is less pronounced (Fig 7a). The productivity is very similar in WARM50 and COLD50 showing that the change in productivity in the SUB20 regions is driven by the local forcings (Fig. 7b). Conversely, the difference in oxygen levels between WARM50 and COLD50 (Fig. 7c) displays similar patterns than between WARM and COLD, highlighting the role of the remote forcings and the change in the

large scale wind patterns, except in the northern part of SUB20, which is mainly governed by the local upwelling processes.

These experiment suggest that the changes in oxygen concentration related with the change from a negative to a positive PDO phase are not directly related to changes in the coastal productivity and

in the upwelling strength, but rather to changes in the large scale circulation in the southern and equatorial part of SUB20.





### 8. Summary of the processes at play and conclusion

We tested here whether the PDO impacts the oxygen concentration in the eastern part of the Pacific
Ocean. We use the forced-ocean model NEMO coupled to a simple NPZD model. After spinup, the
model NEMO-NPZD has been forced by "typical" PDO positive (experiment WARM) and negative
(experiment COLD) conditions derived from the COREv2 atmosphere forcings. A PDO positive
phase is characterized by a decrease of the zonal and meridional wind stress over the Pacific Ocean
by about 5% to 10 %, while the sea surface temperature increases by 0.2°C. A PDO negative event
shows the opposite pattern. In agreement with observations (McPhaden and Zhang, 2003), the
circulation of the tropical Pacific Ocean is more sluggish in the experiment WARM compared to the
experiment COLD by 5 to 10 %. After 50 years of integration, the volume of the suboxic regions
(oxygen lower then 20 mmol.m$^{-3}$) regions is larger by 7 % and the oxygen concentration decreases
by 5 to 50 % in the suboxic regions in WARM compared to COLD.


Recent analyses of the observational datasets showed that the oxygen concentration decreased by
about 5 % in the eastern equatorial Pacific Ocean region since 1960 (Schmidtko et al., 2017; Ito et
al., 2017). We show here that a warm PDO event lasting for 50 years may impact the suboxic region
by at least a similar order of magnitude. The shift to a PDO negative phase (prevailing conditions
before 1975) to a PDO positive phase (prevailing conditions since 1975) may therefore explain a
significant percentage of the large deoxygenation which occurred during the last decades.

The simulated suboxic regions are divided into an equatorial (10°N-10°S), northern (10°N-30°N)
and southern (10°S-30°S) part. The oxygen levels of each sub-region are constrained by different
processes. In the equatorial part, the oxygen levels are set by advective processes (Cabre et al.,
2015; Shigemitsu et al., 2017). In the WARM experiment, the slowing-down of the equatorial
current system (and more particularly of the equatorial undercurrent) decreases the supply of
oxygen (Fig 8). Simultaneously, the supply of nutrients decreases, leading to a decrease in
productivity and respiration. In the eastern part of the basin (in the suboxic regions), the decrease of
supply dominate the change in respiration leading to a net oxygen decrease. Inversely, the change in
respiration are dominant in the mid-Pacific Ocean, highlighting the importance of the
parametrization of the biogeochemical processes (Kriest et al., 2010, Kriest and Oschlies, 2015) and
more particularly the response of the phytoplankton growth to a change in nutrient concentration.





The southern part of the suboxic regions is mostly constrained by isopycnal diffusion processes (Gnanadesikan et al., 2012). In the WARM experiment, the supply of oxygen by isopycnal processes decreases compared to COLD. An hypothesis is the transport by isopycnal processes of less oxygenated water originating from the equatorial regions and the subtropical gyre to the tropics. In the the northern part, the role of diapycnal processes are dominant. In the experiment

WARM, the oxygen decrease is caused by changes in the upwelling system, stimulating productivity and respiration. The relative role of the isopycnal processes is larger in the southern and northern part of the suboxic regions in WARM compared to COLD.

The PDO-related changes are significant at a decadal to multi-decadal time scale, suggesting that

higher frequency climate variability (e.g ENSO) have a weak impact on oxygen levels. The subtropical gyres are slower and equatorward in WARM compared to COLD, which induce an initial increase of oxygen in WARM (increase of diapycnal mixing) in the first years of the spinup in particularly in the southern tropical region (and to a lesser extent in the northern part)

The general slow-down of the large circulation caused by a decrease of the intensity of the trade winds is not reflected in the upwelling strength (Narayan et al., 2010), which is mainly forced by local processes. The strength and seasonality of the upwelling constrain the amount of productivity and then the respiration of oxygen in the suboxic regions. A shift in the upwelling seasonality explains the larger respiration in WARM compared to COLD in the northern part of the suboxic

region. This shift is potentially linked with the NPGO (DiLorenzo et al., 2008), a decadal climate oscillation which signature is implicitly contained into the "typical" PDO forcing. Using 2 supplementary experiments where the WARM and COLD forcings are only applied in the mid Pacific Ocean, we highlighted that a change in the large scale wind pattern constrain the equatorial and southern oxygen levels, while the northern part is constrained by local processes.


One of the largest limitation of our study is that the idealized PDO forcings have been prepared using a relatively short period extending from 1948 to 2007 (the COREv2 dataset). The implicit role of statistically independent climate oscillations such as the NPGO (DiLorenzo et al., 2008) can therefore not be completely ruled out. Another limitation is the resolution of the model. The role of

the mesoscale activity has been previously demonstrated in the supply of oxygen to the suboxic regions of the eastern Pacific Ocean (Montes et al., 2014; Bettencourt et al., 2015; Vergara et al., 2016). The relative increase of the isopycnal diffusive supply in a PDO positive phase suggests an important role of the mesoscale activity. We have shown here that a large part of the changes in





oxygen levels between a typical PDO negative and positive phase are driven by large scale
circulation patterns and include delayed effects, highlighting the difficult trade-off between model
resolution (ideally mesoscale), domain (ideally basin wide), and integration period (ideally multi-
decadal to centennial).

To conclude, our study suggests that the shift from a prolongated (multidecadal) negative to a
prolongated positive PDO phase is accompanied by a decrease in oxygen levels. Several
multidecadal shifts occurred in the last century. For instance, the period 1943-1976 is characterized
by a negative phase, while a warm phase occurred from 1977 to 2011 (Fig 1a). Such shifts occurred
almost every 50 years during the last 1000 years, as shown by PDO reconstructions (MacDonald
and Case, 2005). An open question is whether the succession of these alternate shifts may cause a
change of the oxygen concentration at centennial to millennial time scale.









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




**Figures**

Figure 1 : a – Pacific Decadal Oscillation index (PDO) (annual averages 1900 - 2017) (data from
        the 'Joint Institute for the Study of the Atmosphere and Ocean – University of Washington, USA :
        http://research.jisao.washington.edu/data_sets/pdo). The period 1948 – 2007 has been highlighted.
        The contour line is the smoothed PDO index (20 years running mean). b – average of the zonal 10m
        wind speed (m.s$^{-1}$) of the PDO positive (WARM experiment) phase minus PDO negative phase
(COLD experiment). Contour : zonal wind speed average (m.s$^{-1}$) 1948-2007. c – average of the
        meridional wind speed (m.s$^{-1}$) of the PDO positive phase (WARM experiment) minus PDO negative
        phase (COLD experiment). Contour : meridional wind speed average (m.s$^{-1}$) 1948-2007. d  –
        average of the 10m air temperature (°C) of the PDO positive phase (WARM experiment) minus
        PDO negative phase (COLD experiment). Contour :  average of the 10m air temperature (°C) 1948-
2007. e- difference between the barotropic streamfunction (BSf) WARM and COLD (Sv). Black
        contour: COLD BSf, red contour: WARM BSf. f- difference between the Sea Surface Height (SSH)
        WARM and COLD (m). black contour : SSH COLD. Red contour : SSH WARM. g- difference
        between the meridional overturning (MOC) WARM – COLD (Sv).  black contour : MOC COLD.
        Red contour : MOC WARM. All the differences between WARM and COLD are averaged over 50
years of integration.

        Figure 2 : a – oxygen concentration (mmol.m$^{-3}$) at the end of the spinup (average 100-700m).
        Contour : oxygen concentration of the World Ocean Atlas (WOA) (mmol.m$^{-3}$) (average 100-700m).
        b – thickness (m) of the suboxic regions (oxygen lower than 20 mmol.m$^{-3}$) in WOA and -c at the
end of the spinup. d – difference in percentage between the oxygen concentration in WARM and in
        COLD (average 100-700 m). Black contour : oxygen concentration COLD. Red contour : oxygen
        concentration in WARM (mmol.m$^{-3}$). e – difference between the oxygen concentration (mmol.m$^{-3}$)
        in WARM and in COLD vertically averaged over the suboxic regions SUB20 (defined as the region
        where the oxygen concentration is lower than 20 mmol.m$^{-3}$ at the end of the spinup). Contour :
difference between the oxygen concentrations (mmol.m$^{-3}$) in WARM and COLD (average 100-
        700m). f – difference between the thickness (m) of the suboxic regions in WARM and COLD.  All
        the differences between WARM and COLD are averaged after 50 years of integration.

        Figure 3 : budget of the oxygen concentration (average 100m-700m) (mmol.m$^{-3}$ .yr$^{-1}$) at the end of
the spinup (a-d) and difference between WARM and COLD averaged over 50 years of integration
        (e-f). a,e : respiration. b,f: total supply. c,g : advective processes. d,h: diffusive processes. The
        oxygen concentration (mmol.m$^{-3}$) is displayed in contour in a-d. The oxygen difference between



WARM and COLD (mmol.m$^{-3}$) is displayed in contour in e-f. As a note of caution, a positive value corresponds to a source of oxygen while a negative value corresponds to a sink in a-d, while
differences between two experiments are displayed in e-f.

Figure 4 : a – zonal integration of the vertically averaged oxygen supply in the suboxic regions SUB20 (oxygen lower than 20 mmol.m-3) (mmol.m$^{-2}$.yr$^{-1}$) at the end of the spinup. Black : total supply. Red : advective processes. Blue: isopycnal diffusion. Green : diapycnal diffusion. b –
relative importance of the advective processes (red), isopycnal diffusion (blue) and diapycnal diffusion (green) in the total oxygen supply in SUB20 (Plain : COLD experiment, dashed : WARM experiment). c – zonal integration of the vertically averaged difference of oxygen supply/removal in COLD minus WARM (mmol.m$^{-2}$.yr$^{-1}$) in SUB20. Black: supply. Green: respiration. Red: supply + respiration. d – zonal integration of the vertically averaged difference of oxygen supply in COLD
minus WARM (mmol.m$^{-2.}$ yr$^{-1}$ ) in SUB20. Black : total supply. Red: advective processes. Blue: isopycnal diffusion. Green : diapycnal diffusion.

Figure 5: a – timeseries (50 years) of the difference between WARM and COLD of the oxygen concentration (mmol.m$^{-3}$) in the region EQ (average 10°N-10°S, 160°E-140°W, 100-700m) (black),
SUB20EQ (equatorial part of SUB20: 10°S-10°N) (deep blue), SUB20N (northern part of SUB20: 10°N-30°N) (light blue), SUB20S (southern part of SUB20: 10°S-30°S) (purple). b,c,d : oxygen (mmol.m$^{-3}$) difference (average 100-700 m) between WARM an COLD after b -2 years, c- 10 years, d- 20 years of integration. e-h: timeserie (50 years) of the difference between the WARM and COLD oxygen budget (mmol.m$^{-3}$.yr$^{-1}$) in e-EQ, f-SUB20EQ, g-SUB20N,h-SUB20S. Bold black :
total supply. Blue : advective supply. Pink: diapycnal diffusion. Light blue : dia. + isopycnal diffusion. Bold green : respiration. Bold red: supply + respiration

Figure 6: a - difference (%) of the vertically integrated phytoplankton concentration between WARM and COLD (average of 50 years integration). The vertically integrated phytoplankton
concentration of COLD is shown in contour (mmol.m$^{-2}$). b - difference (%) of the surface phosphate concentration between WARM and COLD (average of 50 years integration). The  surface phosphate concentration of COLD is shown in contour (mmol.m$^{-2}$). c,e,g: average region 10°W-20°S/90°W-60°W and d,f,h region 10°N:20°N/120°W-90°W. c,d:  upwelling (m.yr$^{-1}$) (blue : COLD, red: WARM). e,f : phytoplankton concentration (mmol.m$^{-3}$) (contour : phosphate concentration: mmol.m$^{-3}$) at the end of the spinup. g,h : difference between COLD and WARM in phytoplankton





concentration (mmol.m$^{-3}$) (contour : difference between COLD and WARM in phosphate concentration: mmol.m$^{-3}$)

Figure 7: a – difference between the sea surface height (m) of the experiment WARM50 and
COLD50 (colors) and WARM and COLD (contour). b – difference between the vertically integrated phytoplankton concentration (mmol.m$^{-2}$) of the experiment WARM50 and COLD50 (color) and WARM and COLD (contour). c- difference between the average 100m – 700m oxygen concentration (mmol.m$^{-3}$) of the experiment WARM50 and COLD50 (color) and WARM and COLD (contour).


Figure 8 : summary of the processes at play during a PDO positive (red) and negative phase (blue).





Figure 1





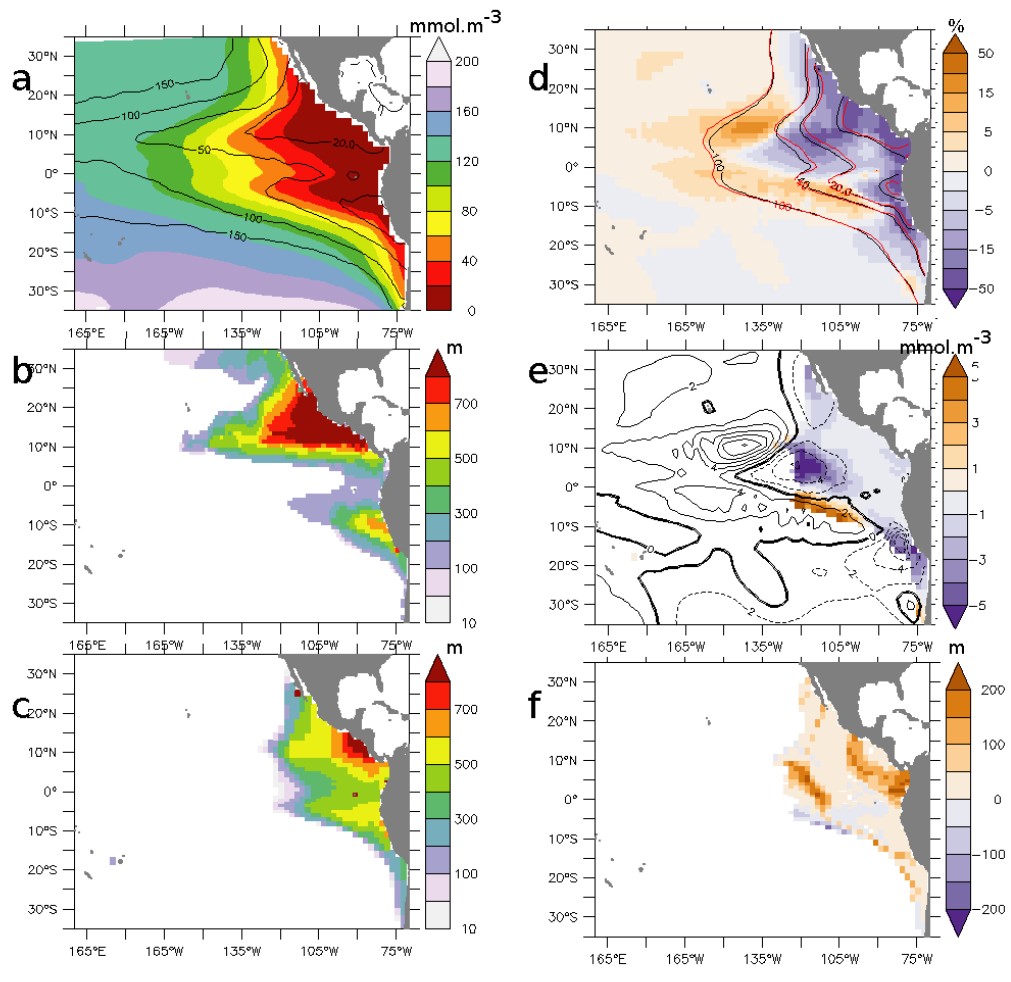

Figure 2





Figure 3





Figure 4



Figure 5





Figure 6



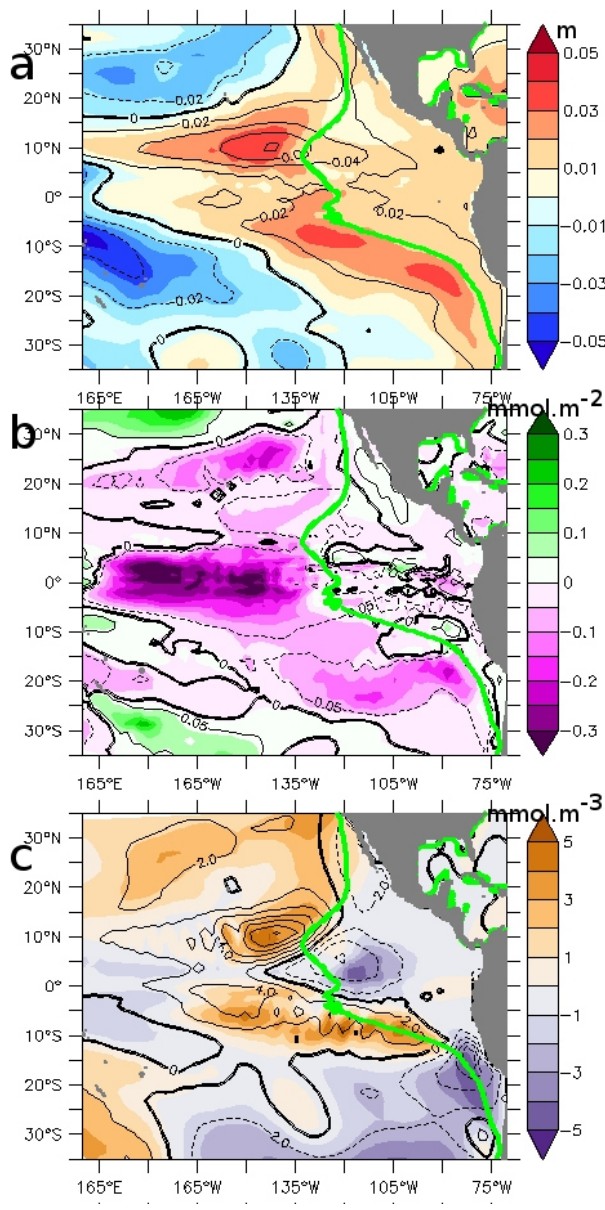

Figure 7



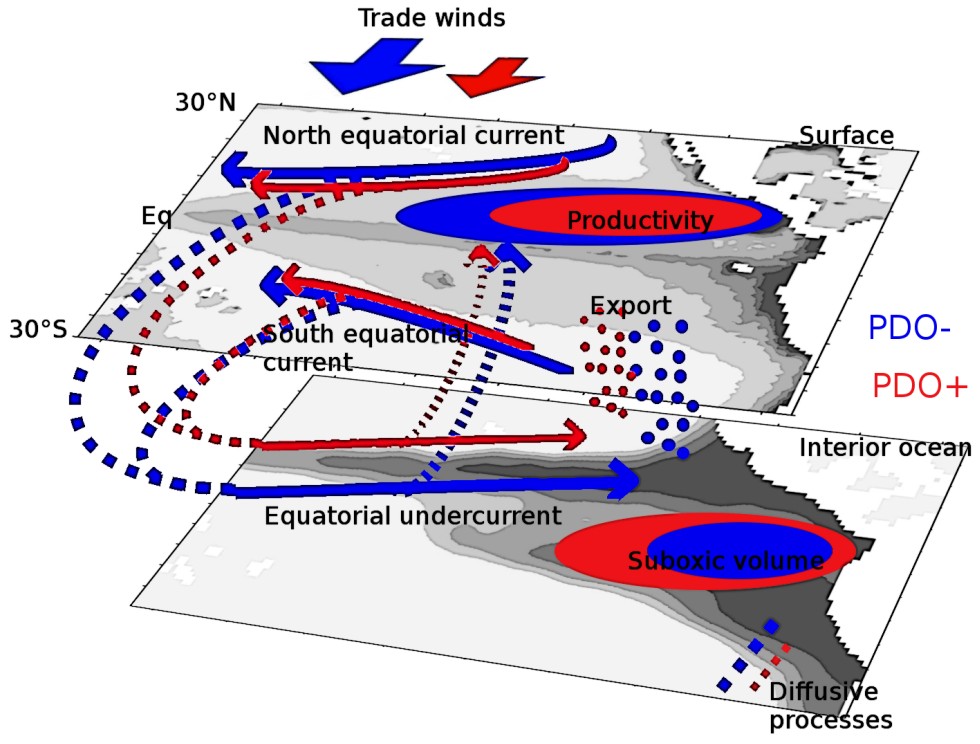

Figure 8