# Peer review of "Pacific Decadal Oscillation and recent oxygen decline in the eastern tropical Pacific Ocean"

_Biogeosciences, 2018_

## Referee Comment (RC1) · Anonymous Referee #1 · 23 Feb 2018

The authors investigate the effect of PDO on the oxygen concentrations in the eastern tropical Pacific. To do this, they used a GCM and carried out several simulations with several atmospheric forcings representing the mean state, PDO positive and negative phases. Their simulation is unique, and the results reveal that the oxygen concentration in the eastern tropical Pacific is decreased due to the shift from the PDO negative to positive phases. The paper is well organized and easy to follow. Below, I list several minor comments for consideration before publication of this paper.

1) The similar paper to theirs has already been published (Deutsch et al., 2011), in which the effect of PDO on oxygen levels in the eastern tropical Pacific was examined using a coarse resolution GCM. They argued that the respiration in the eastern tropical Pacific in the oxygen deficient zone is decreased during the PDO positive phase, and

the dissolved oxygen concentrations tend to increase. The result by Deutsch et al. (2011) seems to be opposite to the result obtained in this study in that the oxygen levels decrease in the PDO positive phase. The authors cited the paper, but did not discuss the difference between their results and those by Deutsch et al. (2011). In this paper, the authors stated that Deutsch et al. (2011) showed that the depth of the thermocline regulates the oxygen levels in the coastal regions of the north eastern subtropical Pacific Ocean. However, to my knowledge, Deutsch et al. (2011) intended to state the effect of PDO on the oxygen levels in the larger spatial scale, i.e., the eastern tropical North Pacific. Thus, the comparison between both results should be needed and the discussion about the comparison is also needed.

2) Lines 130-131: I see that the zonal wind speed in WARM increases compared to MEAN in the mid-equatorial Pacific Ocean in Fig. 1b. Please clarify the point.

3) Line 194: COLD is mistaken for WARM?

4) Figs. 1e and 1g: please define the positive and negative values in the barotropic stream function and the meridional overturning.

5) Lines 684-685: "COLD minus WARM" is "WARM minus COLD"?

6) Line 702: 10°W is 10°S?

7) Line 705: "COLD and WARM" is "WARM and COLD"?

---

## Referee Comment (RC2) · Anonymous Referee #2 · 7 Mar 2018

This paper presents an analysis of idealized experiments conducted with an ocean general circulation model coupled to a simple ocean biogeochemical model. The authors construct idealized forcing representative of positive and negative phases of the PDO, as well as a mean forcing, which is a climatological year from which high-frequency variability has been removed. The difference between the circulation and oxygen fields in the model forced by the warm and cold forcings are suggestive of differences generated by transitions in the PDO. In particular, the warm phase of the PDO is characterized by lower oxygen and larger suboxic volumes than the cold phase. These results suggest that recent declines in may be attributable to transitions in the PDO from cold to warm.

Overall, I find this to be a nice straightforward story and an interesting set of experi-

ments. I think the interpretation of the results is appropriate and well presented. The forcing is highly idealized, but I think the interpretation does not over-reach. I recommend publication pending some very minor revisions.

Minor comments:

ln 14-15: "constrained" is a poor word choice. Perhaps "controlled?"

ln 33: I agree that ocean models capture aspects of this mean picture reasonably well: the overall contrast between high latitude and tropical shadow zones is generally well simulated (in the good models), but the models tend to have OMZs that are much too extensive. In this sense, the model are terrible. This statement needs a bit more nuance.

ln 120-121: I am having trouble parsing this sentence: "The corresponding time steps of the individual annual forcings of the year 1948 – 2007 have been averaged, leading to the reconstruction of a 1 year, 6h temporal resolution, climatological forcing." Please rephrase. I think you are simply saying that you construct an annual climatology of the bandpass-filtered forcing at 6h resolution.

ln 130 and below: the units ms-1 should be m s$^{-1}$

ln 141-142: what's going on here: looks like random text.

---

## Author Comment (AC1) · 4 Jun 2018

We reproduced below the comments of the reviewer in bold and add our replies in black

**Reviewer 1 The authors investigate the effect of PDO on the oxygen concentrations in the eastern tropical Pacific. To do this, they used a GCM and carried out several simulations with several atmospheric forcings representing the mean state, PDO positive and negative phases. Their simulation is unique, and the results reveal that the oxygen concentration in the eastern tropical Pacific is decreased due to the shift from the PDO negative to positive phases. The paper is well organized and easy to follow. Below, I list several minor comments for consideration before publication of this paper**

We thank the reviewer for her/his positive evaluation

**1) The similar paper to theirs has already been published (Deutsch et al., 2011), in which the effect of PDO on oxygen levels in the eastern tropical Pacific was examined using a coarse resolution GCM.**

We agree insofar as that the possible role of the PDO was already pointed out in that previous study (Deutsch et al., 2011). However, while Deutsch et al. build their conclusions on a statistical analysis of a single hindcast experiment driven by realistic forcing that includes all modes of variability, our study exclusively addresses effects of the PDO and addresses its impact on marine oxygen levels by a dedicated set of experiments which allow us to explicitly assess the role of the phase of the PDO on the oxygen levels. The work of Deutsch et al. (2011) is acknowledged in the manuscript and similarities and differences will be discussed in more detail in the revised version.

**They argued that the respiration in the eastern tropical Pacific in the oxygen deficient zone is decreased during the PDO positive phase, and the dissolved oxygen concentrations tend to increase. The result by Deutsch et al. (2011) seems to be opposite to the result obtained in this study in that the oxygen levels decrease in the PDO positive phase. The authors cited the paper, but did not discuss the difference between their results and those by Deutsch et al. (2011). In this paper, the authors stated that Deutsch et al. (2011) showed that the depth of the thermocline regulates the oxygen levels in the coastal regions of the north eastern subtropical Pacific Ocean. However, to my knowledge, Deutsch et al. (2011) intended to state the effect of PDO on the oxygen levels in the larger spatial scale, i.e., the eastern tropical North Pacific.**

Deutsch et al. (2011) indeed focused on the eastern tropical North Pacific. In the revised version we modified the text accordingly:

l531-542: Deutsch et al. (2011, 2014) showed that the depth of the thermocline regulates the oxygen levels in the coastal regions of the north eastern subtropical Pacific

Ocean.

l7269-730: However, we still do not have a clear picture of the impact of the PDO on the suboxic regions and on the oxygen levels of the eastern Pacific Ocean. Indeed, the studies cited above focus either specifically on the suboxic regions of the north eastern tropical Pacific coastal region (Deutsch et al., 2011, 2014) or on the upper thermocline of the tropical Pacific Ocean (Duteil et al., 2014a).

**Thus, the comparison between both results should be needed and the discussion about the comparison is also needed.**

We concur with the reviewer that a more thorough comparison and discussion would be in order, and we have expanded the manuscript accordingly. We added the following text in the discussion L450 :

"Our study can be compared to the study of Deutsch et al. (2011) (thereafter D2011). Based on a 1959-2005 hindcast experiment, D2011 showed that the global suboxic volume (O2 < 5 mmol.m-3), of which 95

The discussion above is completed by a discussion focusing on the limitations of our modelling framework. Using different biogeochemical parameterisations may indeed impact the points (i) and (ii) above and ultimately the oxygen response to a PDO-induced change.

"While our modelling framework captures the general patterns of primary and export production reasonably well, it does not include an iron cycle and may thus, despite displaying a well-tuned mean state, exhibit systematic errors in the sensitivity to environmental changes. Furthermore, the model's representation of the respiration processes is relatively pragmatic. In particular, our model lacks an explicit nitrogen cycle including anaerobic remineralisation by denitrification under low oxygen conditions (Paulmier et al., 2009). Other limitations include for instance a simplistic parameterization of the attenuation of the flux of particulate organic matter that, in our model, neglects any

dependence on temperature or oxygen (Laufkötter et al., 2017). Also not considered in the model is the diel vertical migration of zooplankton also actively transports material into the deep ocean (Bianchi et al., 2015). Anthropogenic activities impact the global biogeochemical cycles. In particular, atmospheric deposition of anthropogenic nitrogen and iron may partially relax the iron limitation in the tropical Pacific Ocean (Ito et al., 2016). Industrial fishing may affect the mortality rate of the zooplankton and possibly feed back on productivity and respiration (Getzlaff and Oschlies, 2017). Each of these 'missing' processes may modulate respiration rates and possibly being correlated with the state of the PDO. An important result of our study is that the PDO-induced changes in respiration are smaller than the PDO-induced changes in oxygen supply by a few percent in most of the eastern tropical Pacific Ocean. We show that this small imbalance integrated for a few decades results in a significant PDO-driven oxygen anomaly that may explain a large part of the observed oxygen decline over the past decades in this region. Experiments including different biogeochemical parameterisations and processes will need to be performed to better assess the robustness of our results."

References

Bianchi, D., Galbraith, E.D., Carozza, D.A., Mislan, K.A.S., Stock, C.A. (2013) Intensification of open-ocean oxygen depletion by vertically migrating animals. Nature Geoscience 6, 545–548.

Getzlaff, J. und Oschlies, A. (2017) Pilot Study on Potential Impacts of Fisheries-Induced Changes in Zooplankton Mortality on Marine Biogeochemistry. Global Biogeochemical Cycles, 31 (11). pp. 1656-1673. DOI 10.1002/2017GB005721.

Ito T., A. Nenes, M. Johnson, N. Meskhidze and C. Deutsch, (2016), ÂăAcceleration of oxygen decline in the tropical Pacific over the past decades by aerosol pollutants, Nature Geosciences,Âădoi:10.1038/ngeo2717

Laufkötter, C., J. G. John, C. A. Stock, and J. P. Dunne (2017), Temperature and oxygen

dependence of the remineralization of organic matter, Global Biogeochem. Cycles, 31, 1038–1050, doi:10.1002/2017GB005643.

Paulmier, A., Kriest, I., and Oschlies, A (2009). Stoichiometries of remineralisation and denitrification in global biogeochemical ocean models, Biogeosciences, 6, 923-935, doi: 0.5194/bg-6-923-2009

**2) Lines 130-131: I see that the zonal wind speed in WARM increases compared to MEAN in the mid-equatorial Pacific Ocean in Fig. 1b. Please clarify the point.**

The sentence L130-131 referring to Fig. 1b is "In WARM the zonal wind speed decreases by about 0.2 to 0.5 ms-1 compared to MEAN in the mid-equatorial Pacific Ocean, where the winds are strongest (at least 8 ms-1) (Fig. 1b). It increases close to the eastern coast by up to 0.3 ms-1, where the winds are weaker (2 to 8 ms-1)"

Figure 1b displays the zonal component of the wind vector with eastward wind velocities having positive values. The zonal wind component is directed westward (negative value). The anomaly WARM – COLD is positive, as the zonal wind component in WARM is "less negative" : the zonal wind speed in WARM decreases (which corresponds to our sentence).

However, the legend of the figure 1 is wrong as "wind speed" should read "wind component (positive eastward)". The original legend of the Fig. 1b is L643-647 is: "b – average of the zonal 10m wind speed (m.s -1 ) of the PDO positive (WARM experiment) phase minus PDO negative phase (COLD experiment). Contour : zonal wind speed average (m.s -1 ) 1948-2007. c – average of the meridional wind speed (m.s -1 ) of the PDO positive phase (WARM experiment) minus PDO negative phase (COLD experiment). Contour : meridional wind speed average (m.s -1 ) 1948-2007."

it should read : "b – average of the zonal 10 m wind component (m.s -1, positive eastward) of the PDO positive (WARM experiment) phase minus PDO negative phase (COLD experiment). Contour : zonal wind component average (m.s -1 ) 1948-2007. c

– average of the meridional wind component (m.s -1, positive northward ) of the PDO positive phase (WARM experiment) minus PDO negative phase (COLD experiment). Contour : meridional wind component average (m.s -1 ) 1948-2007."

**3) Line 194: COLD is mistaken for WARM?**

yes

**4) Figs. 1e and 1g: please define the positive and negative values in the barotropic stream function and the meridional overturning.**

Fig 1e. The sense of rotation is clockwise for positive values Fig 1g. The sense of rotation is clockwise for positive values

**5) Lines 684-685: "COLD minus WARM" is "WARM minus COLD"?**

yes

**6) Line 702: 10 âŮẹ W is 10 âŮẹ S?**

yes

**7) Line 705: "COLD and WARM" is "WARM and COLD"?**

yes

---

## Author Comment (AC2) · 4 Jun 2018

We reproduced below the comments of the reviewer 2 in bold and add our replies in black

**Reviewer 2 - This paper presents an analysis of idealized experiments conducted with an ocean general circulation model coupled to a simple ocean biogeochemical model. The authors construct idealized forcing representative of positive and negative phases of the PDO, as well as a mean forcing, which is a climatological year from which high-frequency variability has been removed. The difference between the circulation and oxygen fields in the model forced by the warm and cold forcings are suggestive of differences generated by transitions**

[Figure]

**in the PDO. In particular, the warm phase of the PDO is characterized by lower oxygen and larger suboxic volumes than the cold phase. These results suggest that recent declines in may be attributable to transitions in the PDO from cold to warm. Overall, I find this to be a nice straightforward story and an interesting set of experiments. I think the interpretation of the results is appropriate and well presented. The forcing is highly idealized, but I think the interpretation does not over-reach. I recommend publication pending some very minor revisions.**

We thank the reviewer for her/his positive evaluation

**Minor comments: In 14-15: "constrained" is a poor word choice. Perhaps "controlled?"**

yes, "controlled" reads much better : "The oxygen levels are mostly controlled by advective processes between 10°N and 10°S"

**In 33: I agree that ocean models capture aspects of this mean picture reasonably well: the overall contrast between high latitude and tropical shadow zones is generally well simulated (in the good models), but the models tend to have OMZs that are much too extensive. In this sense, the model are terrible. This statement needs a bit more nuance.**

Our original sentence L33 is "Ocean models capture this mean picture reasonably well (Bopp et al., 2013; Cocco et al., 2013, Cabré et al., 2015, Shigemitsu et al., 2017)"

We agree with the reviewer that most models tend to have too large OMZs, in particular erroneously including too low oxygen concentrations along the equator close to the eastern boundary. The 'typical biases' of the models are briefly discussed further in the text L177-179: " Typical' biases (Bopp et al., 2013; Cabre et al., 2015) are present in our model. In particular: 1- the OMZ region does not extend far enough westward, in particular north of the equator, 2- oxygen concentrations at the equator close to the eastern boundary are too low".

We replaced the sentence L33 using the more precise phrasing of the reviewer: "The overall contrast between high latitude and tropical shadow zones is generally well reproduced by ocean models (Bopp et al., 2013; Cocco et al., 2013, Cabré et al., 2015, Shigemitsu et al., 2017)".

**In 120-121: I am having trouble parsing this sentence: "The corresponding time steps of the individual annual forcings of the year 1948 – 2007 have been averaged, leading to the reconstruction of a 1 year, 6h temporal resolution, climatological forcing." Please rephrase. I think you are simply saying that you construct an annual climatology of the bandpass-filtered forcing at 6h resolution.**

The sentence has been replaced by : "The corresponding time steps of the individual annual forcings of the years 1948 to 2007 have been averaged to reconstruct a climatological forcing set"

**In 130 and below: the units ms-1 should be m s$-1$**

yes

**In 141-142: what's going on here: looks like random text.**

The paragraph "We spin up the model during 1000 years using the forcing dataset MEAN. We subsequently performed 2 experiments that we integrate for a period of 50 years, which corresponds to the typical oscillation period of the PDO during the past 200 years (Mc Donald and Case, 2005).

L141 -WARM: WARM forcing set (zonal and meridional wind speed, 10m air temperature, 10m humidity)

L142 -COLD : COLD forcing set (zonal and meridional wind speed, 10m air temperature, 10m humidity)"

has been modified :

"We spin up the model over 1000 years using the forcing dataset MEAN. We subsequently performed 2 experiments using the forcing datasets WARM and COLD, respectively, that we both integrate for a period of 50 years, which corresponds to the typical PDO oscillation period during the observational record of the past 200 years (Mc Donald and Case, 2005)"

―――――――――――――――――

---

## Author Response (AR1)

**Reply to review : Pacific Decadal Oscillation and recent oxygen decline in the eastern tropical Pacific Ocean**

Olaf Duteil, Andreas Oschlies , Claus W. Böning

We reproduced below the comments of both reviewers in blue and add our replies in black. The line numbers correspond to the version of the manuscript where the corrections have been highlighted.

Reviewer 1
The authors investigate the effect of PDO on the oxygen concentrations in the eastern tropical Pacific. To do this, they used a GCM and carried out several simulations with several atmospheric forcings representing the mean state, PDO positive and negative phases. Their simulation is unique, and the results reveal that the oxygen concentration in the eastern tropical Pacific is decreased due to the shift from the PDO negative to positive phases. The paper is well organized and easy to follow. Below, I list several minor comments for consideration before publication of this paper.
> We thank the reviewer for her/his positive evaluation

1) The similar paper to theirs has already been published (Deutsch et al., 2011), in which the effect of PDO on oxygen levels in the eastern tropical Pacific was examined using a coarse resolution GCM.
> We agree insofar as that the possible role of the PDO was already pointed out in that previous study (Deutsch et al., 2011). However, while Deutsch et al. build their conclusions on a statistical analysis of a single hindcast experiment driven by realistic forcing that includes all modes of variability, our study exclusively addresses effects of the PDO and addresses its impact on marine oxygen levels by a dedicated set of experiments which allow us to explicitly assess the role of the phase of the PDO on the oxygen levels. The work of Deutsch et al. (2011) is acknowledged in the manuscript and similarities and differences are now discussed in more detail.

They argued that the respiration in the eastern tropical Pacific in the oxygen deficient zone is decreased during the PDO positive phase, and the dissolved oxygen concentrations tend to increase. The result by Deutsch et al. (2011) seems to be opposite to the result obtained in this study in that the oxygen levels decrease in the PDO positive phase. The authors cited the paper, but did not discuss the difference between their results and those by Deutsch et al. (2011). In this paper, the authors stated that Deutsch et al. (2011) showed that the depth of the thermocline regulates the oxygen levels in the coastal regions of the north eastern subtropical Pacific Ocean. However, to my knowledge, Deutsch et al. (2011) intended to state the effect of PDO on the oxygen levels in the larger spatial scale, i.e., the eastern tropical North Pacific.

Deutsch et al. (2011) indeed focused on the eastern tropical North Pacific. In the revised version we modified the text accordingly:
l47: Deutsch et al. (2011, 2014) showed that the depth of the thermocline regulates the oxygen levels in the  north eastern subtropical Pacific Ocean.
L64-65: However, we still do not have a clear picture of the impact of the PDO on the suboxic regions and on the oxygen levels of the eastern Pacific Ocean. Indeed, the studies cited above focus either specifically on the suboxic regions of the north eastern tropical Pacific  region (Deutsch et al., 2011, 2014) or on the upper thermocline of the tropical Pacific Ocean (Duteil et al., 2014a).

Thus, the comparison between both results should be needed and the discussion about the comparison is also needed.

> We concur with the reviewer that a more thorough comparison and discussion would be in order, and we have expanded the manuscript accordingly. We added the following text in the discussion L436 :

"Our study can be compared to the study of Deutsch et al. (2011) (thereafter D2011). Based on a 1959-2005 hindcast experiment, D2011 showed that the global suboxic volume (O2 < 5 mmol.m-3), of which 95 % is contained in the north eastern tropical Pacific Ocean (0° to 30°N, 140°W to the coast), is controlled by the depth of the thermocline that constrains the productivity and the amount of oxygen respired around the suboxic volume. The depth of the thermocline is strongly related to the strength of the PDO, which explains 24 % of the variability of the suboxic volume in the hindcast simulation of D2011. In D2011, a PDO-negative phase was characterized by a large extent of the suboxic regions resulting from enhanced respiration with only negligible effects of changes in oxygen supply. Our results differ from D2011 because the increase of the respiration in a PDO-negative phase (higher primary production) is more than compensated by an increase in oxygen supply by advective / diffusive processes, leading ultimately to elevated oxygen levels and smaller suboxic regions in a PDO-negative state. Considering that the models used by D2011 and in our study are similar with similar grid resolution, the balance between changes in oxygen consumption (dominant effect in D2011) and transport (dominant effect in our study) depends on (i) the response of primary productivity and export production to a nutrient increase, and (ii) the depth of the suboxic regions, as respiration changes are stronger, and more difficult to compensate by oxygen supply changes, at shallower than at greater depths. While the biogeochemical model used in our study contains fully prognostic nutrient, phytoplankton, zooplankton and detritus fields, D2011 used a simple restoring model that diagnosed export production from restoring simulated against observed surface phosphate concentrations. Thereby, the model of D2011 did not account for possible PDO-driven changes in surface nutrient concentrations and instead likely overestimated the variability in export and respiration at depth"

The discussion above is completed by a discussion focusing on the limitations of our modelling framework. Using different biogeochemical parameterisations may indeed impact the points (i) and (ii) above and ultimately the oxygen response to a PDO-induced change.

"While our modelling framework captures the general patterns of primary and export production reasonably well, it does not include an iron cycle and may thus, despite displaying a well-tuned mean state, exhibit systematic errors in the sensitivity to environmental changes. Furthermore, the model's representation of the respiration processes is relatively pragmatic. In particular, our model lacks an explicit nitrogen cycle including anaerobic remineralisation by denitrification under low oxygen conditions (Paulmier et al., 2009). Other limitations include for instance a simplistic parameterization of the attenuation of the flux of particulate organic matter that, in our model, neglects any dependence on temperature or oxygen (Laufkötter et al., 2017). Also not considered in the model is the diel vertical migration of zooplankton also actively transports material into the deep ocean (Bianchi et al., 2015). Anthropogenic activities impact the global biogeochemical cycles. In particular, atmospheric deposition of anthropogenic nitrogen and iron may partially relax the iron limitation in the tropical Pacific Ocean (Ito et al., 2016). Industrial fishing may affect the mortality rate of the zooplankton and possibly feed back on productivity and respiration (Getzlaff and Oschlies, 2017). Each of these 'missing' processes may modulate respiration rates and possibly being correlated with the state of the PDO. An important result of our study is that the PDO-induced changes in respiration are smaller than the PDO-induced changes in oxygen supply by a few percent in most of the eastern tropical Pacific Ocean. We show that this small imbalance integrated over a few decades results in a significant PDO-driven oxygen anomaly that may explain a large part of the observed oxygen decline over the past decades in this region. Experiments including different biogeochemical parameterisations and processes will need to be performed to better assess the robustness of our results."

Figure 1b displays the zonal component of the wind vector with eastward wind velocities having positive values. The zonal wind component is directed westward (negative value). The anomaly WARM – COLD is positive, as the zonal wind component in WARM is "less negative" : the zonal wind speed in WARM decreases (which corresponds to our sentence).

However, the legend of the figure 1 was wrong as "wind speed" should have read "wind component (positive eastward)". The original legend of the Fig. 1b was "b – average of the zonal 10m wind speed (m.s -1) of the PDO positive (WARM experiment) phase minus PDO negative phase (COLD experiment). Contour : zonal wind speed average (m.s -1) 1948-2007. c – average of the meridional wind speed (m.s -1) of the PDO positive phase (WARM experiment) minus PDO negative phase (COLD experiment). Contour : meridional wind speed average (m.s -1) 1948-2007."

The name of the experiments have been changed for clarity reasons : WARM is now PDO_PLUS and COLD is now PDO_MINUS. The sentence now reads:
"b – average of the zonal 10 m wind component (m.s $^{-1}$, positive eastward) of the PDO positive (PDO_PLUS experiment) phase minus PDO negative phase (PDO_MINUS experiment). Contour : zonal wind component average (m.s $^{-1}$) 1948-2007. c – average of the meridional wind component (m.s $^{-1}$, positive northward ) of the PDO positive phase (PDO_PLUS experiment) minus PDO negative phase (PDO_MINUS experiment). Contour : meridional wind component average (m.s $^{-1}$) 1948-2007."

3) Line 194: COLD is mistaken for WARM?
> yes. Many thanks for pointing out this mistake!

4) Figs. 1e and 1g: please define the positive and negative values in the barotropic stream function and the meridional overturning.

> Fig 1e. The sense of rotation is clockwise for positive values
> Fig 1g. The sense of rotation is clockwise for positive values

5) Lines 684-685: "COLD minus WARM" is "WARM minus COLD"?
> yes. Many thanks for pointing out this mistake!

6) Line 702: 10 ◦ W is 10 ◦ S?
> yes. Many thanks for pointing out this mistake!

7) Line 705: "COLD and WARM" is "WARM and COLD"?
> yes
* * *
Reviewer 2
This paper presents an analysis of idealized experiments conducted with an ocean general circulation model coupled to a simple ocean biogeochemical model. The authors construct idealized forcing representative of positive and negative phases of the PDO, as well as a mean forcing, which is a climatological year from which high-frequency variability has been removed. The difference between the circulation and oxygen fields in the model forced by the warm and cold forcings are suggestive of differences generated by transitions in the PDO. In particular, the warm phase of the PDO is characterized by lower oxygen and larger suboxic volumes than the cold phase. These results suggest that recent declines in may be attributable to transitions in the PDO from cold to warm. Overall, I find this to be a nice straightforward story and an interesting set of experiments. I think the interpretation of the results is appropriate and well presented. The forcing is highly idealized, but I think the interpretation does not over-reach. I recommend publication pending some very minor revisions.
> We thank the reviewer for her/his positive evaluation

Minor comments:
ln 14-15: "constrained" is a poor word choice. Perhaps "controlled?"
> yes, "controlled" reads much better : "The oxygen levels are mostly controlled by advective processes between 10°N and 10°S"

ln 33: I agree that ocean models capture aspects of this mean picture reasonably well: the overall contrast between high latitude and tropical shadow zones is generally well simulated (in the good models), but the models tend to have OMZs that are much too extensive. In this sense, the model are terrible. This statement needs a bit more nuance.
> Our original sentence L33 (L29) is "Ocean models capture this mean picture reasonably well (Bopp et al., 2013; Cocco et al., 2013, Cabré et al., 2015, Shigemitsu et al., 2017)"

We agree with the reviewer that most models tend to have too large OMZs, in particular erroneously including too low oxygen concentrations along the equator close to the eastern boundary. The 'typical biases' of the models are briefly discussed further in the text L156: " Typical' biases (Bopp et al., 2013; Cabre et al., 2015) are present in our model. In particular: 1- the OMZ region does not extend far enough westward, in particular north of the equator, 2- oxygen concentrations at the equator close to the eastern boundary are too low".

We replaced the sentence L29 using the more precise phrasing of the reviewer: "The overall contrast between high latitude and tropical shadow zones is generally well reproduced by ocean models (Bopp et al., 2013; Cocco et al., 2013, Cabré et al., 2015, Shigemitsu et al., 2017)".

ln 120-121: I am having trouble parsing this sentence: "The corresponding time steps of the individual annual forcings of the year 1948 – 2007 have been averaged, leading to the reconstruction of a 1 year, 6h temporal resolution, climatological forcing." Please rephrase. I think you are simply saying that you construct an annual climatology of the bandpass-filtered forcing at 6h resolution.
> The sentence L121 (L112) has been replaced by : "The corresponding time steps of the individual annual forcings of the years 1948 to 2007 have been averaged to reconstruct a climatological forcing set"

ln 130 and below: the units ms-1 should be m s$^{-1}$
> yes

ln 141-142: what's going on here: looks like random text.
> The paragraph "We spin up the model during 1000 years using the forcing dataset MEAN. We subsequently performed 2 experiments that we integrate for a period of 50 years, which corresponds to the typical oscillation period of the PDO during the past 200 years (Mc Donald and Case, 2005).
-WARM: WARM forcing set (zonal and meridional wind speed, 10m air temperature, 10m humidity)
-COLD : COLD forcing set (zonal and meridional wind speed, 10m air temperature, 10m humidity)"

has been modified (L126):
"We spin up the model over 1000 years using the forcing dataset MEAN. We subsequently performed 2 experiments using the forcing datasets PDO_PLUS and PDO_MINUS, respectively, that we both integrate for a period of 50 years,

which corresponds to the typical PDO oscillation period during the observational record of the past 200 years (Mc Donald and Case, 2005)"
* * *
Additional corrections beyond those requested by the reviewers :

- The name of the experiments have been changed for clarity reasons : WARM is now PDO_PLUS and COLD is now PDO_MINUS

- Also for clarity reasons we rephrased the part 7.2. The experiments and their interpretation are unchanged, but the presentation should have been improved. The original and new versions of the part 7.2 are displayed below.

[revised manuscript text omitted]

- The following paragraphs in the discussion have been deleted as they are either redundant or vague.
L400: "The PDO-related changes are significant at a decadal to multi-decadal time scale, suggesting that higher frequency climate variability (e.g ENSO) have a weak impact on oxygen levels. The subtropical gyres are slower and

equatorward in WARM compared to COLD, which induce an initial increase of oxygen in WARM (increase of diapycnal mixing) in the first years of the spinup in particularly in the southern tropical region (and to a lesser extent in the northern part)"

[revised manuscript text omitted]

Figure 1

[Figure]

Figure 2

[Figure]

Figure 3

[Figure]

Figure 4

[Figure]

Figure 5

[Figure]

Figure 6

[Figure]

Figure 7

[Figure]

Figure 8

---

## Author Response (AR2)

**Pacific Decadal Oscillation and recent oxygen decline in the eastern tropical Pacific Ocean**

Duteil et al. - minor revision

The comments of the editor are reproduced below **in bold**. We reply in thin.

**Page 1, line11: expand and not expend**
Corrected

**Lines 40-42: Please reformulate making clear that these are the questions that you will address in this paper. For instance, in the present work, we aim to determine whether the shift from …..**
The text L40-42 is "Could the shift from a negative phase of the PDO (prevailing conditions before 1975) to a positive phase (prevailing conditions since 1975) partly explain the current deoxygenation occurring in the eastern tropical Pacific Ocean ? Which mechanisms are at play and in which regions ?". We deleted this paragraph as the goal of this study is detailed L80-85.

**Lines 46-60: This part is really difficult to follow, especially for non-expert of the Eastern Pacific circulation and thermocline variability. I would suggest that you make it more pedagogical. For instance, a figure summarizing the main mechanisms at play would be really helpful.**

The text L46-60 is "The stronger trade winds occurring during a negative phase of the PDO cause a shoaling of the eastern thermocline of the tropical and subtropical Pacific Ocean (Miller et al., 1994). Deutsch et al. (2011, 2014) showed that the depth of thethermocline regulates the oxygen levels in the north eastern subtropical Pacific Ocean. In these regions, a shallower thermocline fosters low oxygen concentrations in the intermediate ocean as a larger amount of organic material is respired below the mixed layer (Deutsch et al., 2011).

Simultaneously, in the tropical regions the zonal volume transport by the equatorial current system increases during a negative PDO event as does the meridional transport by the sub-tropical cells (STCs) (Hong et al., 2014), which connect the subtropics to the tropics (McCreary and Lu, 1994). The variability of the strength of the STCs forces the variability of the oxygen transport in the upper thermocline of the equatorial Pacific Ocean (Duteil et al., 2014a). A competition takes place between the increased oxygen transport and the increased respiration as primary production is enhanced by the increased nutrient supply.

Finally, stronger trade winds also increase the subduction volume of the North (Qu et al., 2009) and South Pacific Eastern Subtropical Mode Waters (Luo et al., 2011). A subtropical increase of productivity related to an increase of the trade winds causes a negative oxygen anomaly in these mode waters, which is transported equatorward and leads to a delayed oxygen decrease in tropical regions as shown by Ridder and England (2014) in an Earth System Model of intermediate complexity."

We added a reference to the figure below (Supplementary Figure 1) :
L46 "see the Supplementary Figure 1 for an overview of the mechanisms controlling the oxygen levels in the eastern tropical ocean"

[Figure]

Supplementary figure 1 : overview of the mechanisms controlling the oxygen levels in the tropical Pacific ocean (the north tropical region, from the equator to approximately 30°N, is displayed here for clarity reasons). The upper part of the thermocline is displayed in red.

1. "trade winds", playing a major role in the control of the strength of the ocean currents in the tropical / subtropical regions. 2. water subduction in the subtropical gyres. The water masses are transported toward the eastern tropical regions by 3. the subtropical-tropical cells and 4. the equatorial current system. 5. strong primary productivity in the eastern part of the basin. 6. strong export productivity and sluggish circulation, fostering the existence of oxygen minimum zones (figure adapted from O.Duteil, A. Oschlies, C.Böning, M.Scheinert, GEOMAR, 2014, pers. comm.)

[revised manuscript text omitted]

Figure 1

[Figure]

Figure 2

[Figure]

Figure 3

[Figure]

Figure 4

[Figure]

Figure 5

[Figure]

Figure 6

[Figure]

Figure 7

[Figure]

Figure 8

**Overview of the mechanisms controlling the oxygen levels in the tropical Pacific Ocean**

[Figure]

Supplementary figure 1 : overview of the mechanisms controlling the oxygen levels in the tropical Pacific ocean (the north tropical region, from the equator to approximately 30°N, is displayed here for clarity reasons). The upper part of the thermocline is displayed in red.
1. "trade winds", playing a major role in the control of the strength of the ocean currents in the tropical / subtropical regions. 2. water subduction in the subtropical gyres. The water masses are transported toward the eastern tropical regions by 3. the subtropical-tropical cells and 4. the equatorial current system. 5. strong primary productivity in the eastern part of the basin. 6. strong export productivity and sluggish circulation, fostering the existence of oxygen minimum zones (figure adapted from O.Duteil, A.Oschlies, C.Böning, M.Scheinert, GEOMAR, 2014, pers. comm.)